# Academic case reports lack diversity: Assessing the presence and diversity of sociodemographic and behavioral factors related to Post COVID-19 Condition

Juan Andres Medina Florez[1], Shaina Raza[3], Rashida Lynn Ansell[2], Zahra Shakeri[1], Brendan T. Smith[1,4], Elham Dolatabadi[1,2,3]*

1 Dalla Lana School of Public Health, University of Toronto, Toronto, Canada, 2 York University, Toronto, Canada, 3 Vector Institute, Toronto, Canada, 4 Public Health Ontario, Toronto, Canada

* edolatab@yorku.ca

**Data availability statement:** We have uploaded our data to Zenodo. It is accessible via the

## Abstract

Understanding disparities in the prevalence of Post COVID-19 Condition (PCC) amongst vulnerable populations is crucial to improving care and addressing intersecting inequities. This study aims to develop a comprehensive framework for integrating social determinants of health (SDOH) into PCC research by leveraging natural language processing (NLP) techniques to analyze disparities and variations in SDOH representation within PCC case reports. Following construction of a PCC Case Report Corpus, comprising over 7,000 case reports from the LitCOVID repository, a subset of 709 reports were annotated with 26 core SDOH-related entity types using pre-trained named entity recognition (NER) models, human review, and data augmentation to improve quality, diversity and representation of entity types. An NLP pipeline integrating NER, natural language inference (NLI), trigram and frequency analyses was developed to extract and analyze these entities. Both encoder-only transformer models and RNN-based models were assessed for the NER objective.

Fine-tuned encoder-only BERT models outperformed traditional RNN-based models in generalizability to distinct sentence structures and greater class sparsity, achieving a macro F1-score of 0.72 and macro AUC of 0.99 on a held-out generalization set. Exploratory analysis revealed variability in entity richness, with prevalent entities like condition, age, and access to care, and under-representation of sensitive categories like race and housing status. Trigram analysis highlighted frequent co-occurrences among entities, including age, gender, and condition. The NLI objective (entailment and contradiction analysis) showed attributes like "Experienced violence or abuse" and "Has medical insurance" had high entailment rates (82.4%–80.3%), while attributes such as "Is female-identifying," "Is married," and "Has a terminal condition" exhibited high contradiction rates (70.8%–98.5%).

Our results highlight the effectiveness of transformer-based NER in extracting SDOH information from case reports. However, the findings also expose critical gaps in the

following link: https://doi.org/10.5281/zenodo.15668020. Importantly, this includes a file with all annotated case reports in CoNLL formal, in addition to a file with all the associated case reports sections.

**Funding:** Resources used in preparing this research were provided, in part, by the Province of Ontario, the Government of Canada through CIFAR, and companies sponsoring the Vector Institute www.vectorinstitute.ai/partnerships/. This publication was supported by the Canadian Institutes of Health Research (CIHR), Funding Reference Number 192124. The CAN-TAP-TALENT is funded by the Canadian Institutes of Health Research (CIHR) – FRN 184898. The authors wish to acknowledge the CAN-TAP-TALENT for its role in supporting the completion of this CAN-TAP-TALENT Research Project. The funders had no role in study design, data collection and analysis, decision to publish, or preparation of the manuscript.

**Competing interests:** The authors declare that they have no known competing financial interests or personal relationships that could have appeared to influence the work reported in this paper.

representation of marginalized groups within PCC-related academic case reports, e.g., across gender, insurance status, and age. This work underscores the need for standardized SDOH documentation and inclusive reporting practices to enable more equitable research and inform future health policy and AI model development.

# 1 Introduction

The acute phase of the COVID-19 pandemic may have passed, but its long-term impacts continue to affect millions worldwide [1–3]. According to the World Health Organization (WHO) [4], 10–20% of COVID-19 cases lead to post-COVID condition (PCC), also known as long COVID. The Centers for Disease Control and Prevention (CDC) [5] reports 6.9% of U.S. adults have experienced PCC, with 3.4% still symptomatic. A November 2024 analysis by United Press International (UPI) [6] estimates that 23% of Americans who had COVID-19 may experience symptoms of PCC, highlighting the substantial and ongoing public health challenges posed by PCC.

While significant research efforts have been deployed to understand and mitigate PCC, from identifying symptoms and improving clinical care to characterizing its biological underpinnings, there remains a critical and underexplored area: the role of social and behavioral factors in shaping who gets PCC, how they experience it, and what care they receive [7–11]. Structural inequities rooted in race, income, gender identity, and access to healthcare are known to influence health outcomes broadly [12–14]. In the context of PCC, early evidence suggests that these factors may also play a key role in disparities related to symptom severity, duration, and access to treatment. For instance, studies have identified the disproportionate prevalence of PCC among certain population subgroups, including females, transgender individuals, and Hispanics [15–19]. Importantly, as scientific knowledge of PCC continues to expand and deepen [20], a significant knowledge gap persists concerning why disparities in PCC prevalence amongst subgroups exist or how to reduce them.

A major barrier to this understanding is the lack of comprehensive datasets that capture both sociodemographic and behavioral variables, such as race, ethnicity, age, income, and occupational status, and PCC [21,22]. For instance, structured clinical Datasets, such as Electronic or Medical Health Records often fail to include detailed sociodemographic variables, limiting analyses of how systemic inequities impact PCC outcomes, such as unequal access to healthcare and its impact on PCC outcomes [23–26]. Research has highlighted the inadequacy of race and ethnicity data in electronic health records, an issue compounded by inconsistent collection and reporting practices [26–28].

The scarcity of such data hinders efforts to identify actionable pathways for disparity reduction. Bridging these gaps requires advancements in data collection and integration. Furthermore, the integration of additional data sources, such as clinical case reports, could complement structured datasets, offering richer insights into patients' experiences and social contexts. By combining these reports with advanced natural language processing (NLP) techniques—such as entity extraction [29], contextual understanding [30], language entailment [31], and classification [32]—researchers can uncover complex relationships between sociodemographic factors and PCC development [33,34].

The overarching objective of this study is to establish an approach for advancing the integration of SDOH into PCC research through state-of-the-art NLP methodologies. In particular, this research is guided by the central question: How are social determinants of health represented in academic case reports of PCC, and what gaps or disparities exist in their reporting? By addressing this question, we aim to generate foundational insights into how SDOH

are documented in the scientific literature on PCC and to evaluate the strengths and limitations of case reports as a data source for equity-focused health research. Our experiments demonstrate that fine-tuned BERT-based models outperform traditional RNN architectures in extracting SDOH-related entities. In addition, our analysis revealed contradictions in the reporting of attributes such as gender identity and insurance status, indicating potential systemic biases and omissions within the academic case report literature. This study makes three key contributions to the field:

1. **Corpus Creation:** We release an open-source, novel PCC Case Report Corpus comprising more than 7,000 case reports, annotated using the proposed fine-tuned BERT-base-uncased model. This corpus includes 26 sociodemographic, behavioral, and clinical entities such as age, vaccination status and medical condition, along with a robust data processing pipeline designed to improve reproducibility and facilitate future research in PCC. We are releasing our dataset to the research community for reproducibility of the experiments. The dataset is hosted on Zenodo and is accessible via the following link: https://doi.org/10.5281/zenodo.15668020.

2. **Pipeline Development:** We introduce a comprehensive end-to-end NLP pipeline that integrates named entity recognition (NER) and natural language inference (NLI) techniques, enabling the extraction and entailment of entities from case reports. This pipeline utilizes a combination of data augmentation, regularization, rule-based methods and generative AI, and categorizes the extracted entities into meaningful groupings, providing a structured framework for further analysis.

3. **Gap Analysis:** We identify key gaps in the representation of SDOH attributes in PCC, such as under-representation of race and spiritual beliefs, and reveal patterns of agreement and contradictions in entity co-occurrences, such as disparities between documented insurance coverage and access to care.

In doing so, we seek to inform improvements in data comprehensiveness and equity, ultimately guiding more inclusive and effective research and policy development for PCC.

The remainder of the paper is organized as follows. Sect 2 reviews related work on NLP applications in clinical and SDOH applications. Sect 3 outlines the methodology. Sect 4 presents the results from model benchmarking, entity distribution analysis, and NLI-based findings. Sect 5 discusses key findings, implications, and limitations. Sect 6 concludes the paper with a summary of contributions and directions for future work.

## 2 Related work

Previous efforts utilizing a named entity recognition algorithm to extract social determinants of health from textual data span various health domains, with diverse SDOH entity dimensions; however, these efforts remain limited [35]. Most frequently, the corpora utilized for model development hails from sources such as electronic health records (EHRs), clinical narratives, and academic research reports [34]. For instance, Lituiev et al. (2023) utilize ROBERta and a hybrid transformer model to extract 10 entity domains (7 SDOH domains) from medical notes within the context of chronic lower-back pain. Importantly, their selected SDOH domains are limited to social support, marital status, finance, food, transportation, housing, and insurance [36]. Yu et al. (2024), on the other hand, compared a range of 7 transformer models to extract over 20 SDOH entity domains in clinical narratives related to cancer and opioid use, including domains related to economic stability, education, health care, community context, physical environment, gender and race [37]. Most closely aligned with the

objective of this analysis, previous work by Bashir et al. (2022) and Raza et al. (2023) utilize a BiLSTM-CNN-CRF model and a hybrid BioBERT model, respectively, for COVID-19-related SDOH entity extraction from clinical case reports [12,38]. Notably, Bashir et al. (2022) extract entities relating to over 16 sociodemographic entity domains including patient name, date, location, employment, gender, and sexual orientation, to name a few, in addition to a range of clinical entities relevant to COVID-19 [38]. Raza et al. (2023), on the other hand, extract SDOH entities including sociodemographic, biometric, temporal, lifestyle, socioeconomic, and healthcare system-related factors, in addition to clinical entities such as cardiovascular, metabolic, renal, oncological, respiratory, and psychological conditions, to name a few [12].

While the utilization of NLP in the context of PCC is not novel, these efforts have been largely focused on the identification of clinical and biological entities, such as symptoms, diagnoses, and treatments [3,39–43]. To our knowledge, NLP techniques have not been specifically tailored to extract a comprehensive set of sociodemographic entities in the PCC context. As a result, this study is unique in that it entails the development of a named entity recognition tool to extract sociodemographic and behavioral entities within the PCC context. Moreover, it also provides a novel utilization of a language entailment framework to extract insights from labeled data within the PCC context.

## 3 Materials and methods

### Ethics statement

This study exclusively utilized de-identified, publicly available case reports from the Lit-COVID. (National Institute of Health. https://www.ncbi.nlm.nih.gov/research/coronavirus). The dataset is made available under the Open Database License (ODbL). As no human subjects were directly involved and no identifiable private information was accessed, this research did not require institutional ethics approval.

### Corpus construction for PCC case reports

We begin by describing the construction of our dataset, which forms the foundation for all analyses. The PCC Case Report Corpus was developed by sourcing relevant case report articles from the LitCOVID repository [44] guided by a query incorporating the keywords "Post COVID," "Long COVID," and "Post-acute COVID-19 syndrome." Inclusion criteria specified case reports published from January 1, 2020, to October 16, 2023, in English, featuring patients with a confirmed history of SARS-CoV-2 infection and documented PCC symptoms or complications. Only full-text articles were considered. Exclusion criteria included preprints, non peer-reviewed articles, review articles, meta-analyses, systematic reviews, studies focused exclusively on acute COVID-19, articles lacking detailed clinical information, and non-human studies.

Approximately 10,000 papers meeting these criteria were retrieved from LitCOVID. PDFs of the relevant articles were converted into images, processed with optical character recognition (OCR), and structured using the John Snow Labs Health OCR tool [45] (see Fig 1 for the full pipeline). To minimize background noise, the case report sections were extracted from each document, as preliminary exploration showed that these sections contained most of the socioeconomic and clinical attributes relevant to the subjects of interest. Non-relevant content was systematically excluded using rule-based regular expressions, such that documents lacking case report sections were removed, and rules were iteratively refined based on observed errors. This process yielded a curated corpus of 7,172 case reports for further analysis. Three

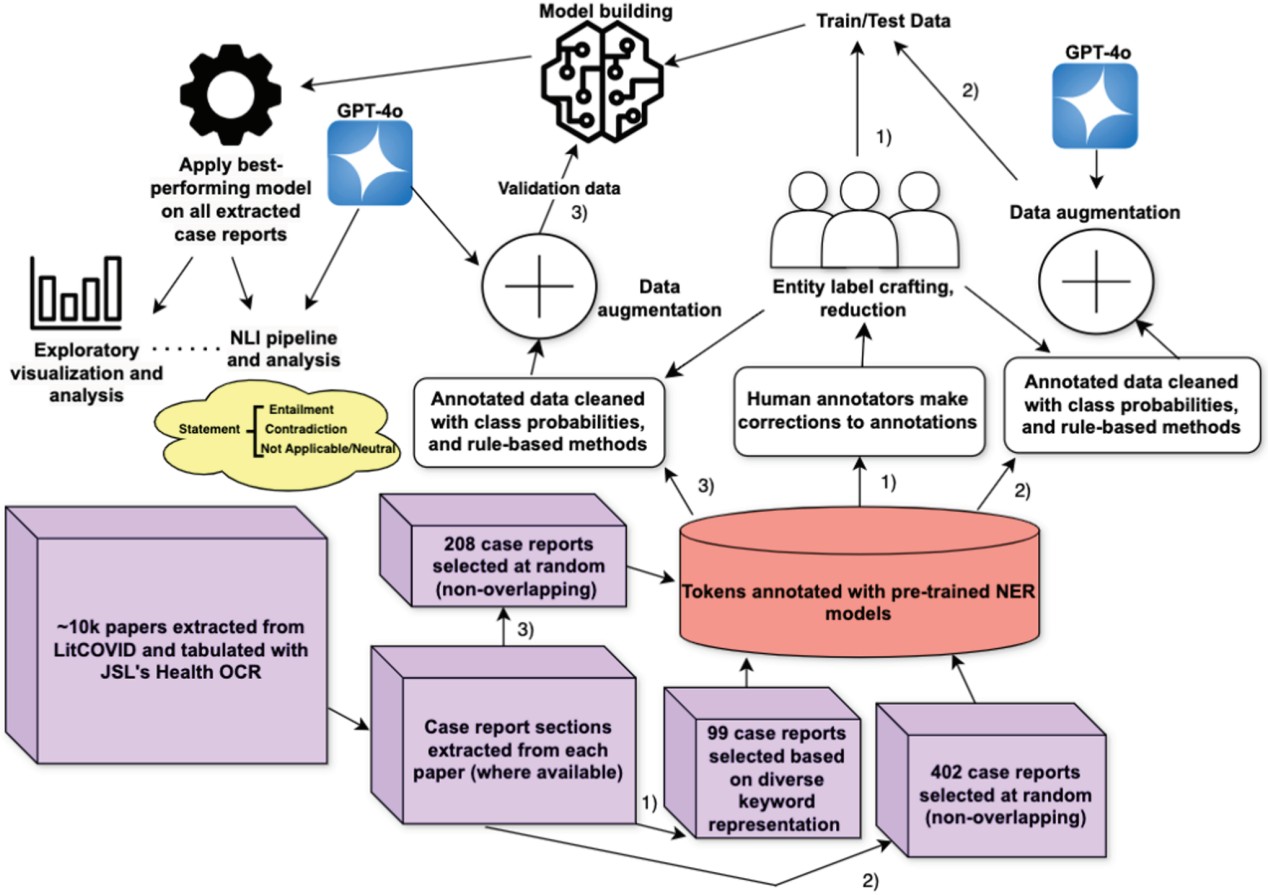

**Fig 1. The PCC Case Report Corpus was developed from LitCOVID case reports (January 2020–October 2023) featuring post-COVID symptoms, using keyword-based searches and strict inclusion criteria.** After filtering and processing, 7,172 case reports were curated, with 709 selected for NER model development: 99 in subset 1 (keyword-filtered for diversity), 402 in subset 2, and 208 in subset 3 (randomly selected). Annotation involved initial NER, human review, and data augmentation, balancing quality and efficiency to enhance the model's ability to capture PCC complexities.

non-overlapping subsets, totaling 709 case reports, were selected for NER model development. These are referred to as subset 1, with 99 case reports, subset 2, with 402 case reports, and subset 3, with 208 case reports, as shown in Fig 1. Of these, subsets 1 and 2, totaling 501 case reports, were used for model training, with an 80/20 random split applied to create a training set and an optimization testing set. The remaining 208 case reports (subset 3) were reserved for evaluating model generalization. The size of each subset was determined based on the feasibility of processing and annotation, while ensuring that the sizes remained relatively proportionate to each other based on their end purpose (development vs. validation set). In this study, the 20% subset from the training data is referred to as the optimization testing set, while the 208 reserved case reports are referred to as the generalization evaluation set.

The 99 case reports utilized for subset 1 were selected with a keyword-based and rule-based filtration process to ensure that there would be a representation of a variety of labels and that overall text quality would be adequate. In particular, keywords representing marginalized and at-risk populations were prioritized to ensure diversity in our training data,

including homelessness, housing-insecurity, low-income, poverty, black, hispanic, uninsured, abused, bisexual, homosexual, and female. Subsets 2 and 3, on the other hand, were selected at random from non-overlapping subsets of the dataset. Additional initial explorations on the corpus are outlined in supplementary materials.

## Annotation process

With the dataset in place, we developed an annotation strategy to label key sociodemographic, behavioral, and clinical entities. The annotation process consisted of four main steps of initial entity extraction using pre-trained NER models by John Snow Labs (JSL), human review, entity refinement and a combination of data filtration and augmentation. The entire dataset of 709 case reports underwent the initial entity extraction step, while steps of human review, and data filtration and augmentation were applied to targeted subsets 1 and 2–3, respectively. This tiered approach addressed two key challenges: first, that full manual annotation of all 709 reports would be resource-intensive, and second, that the use of a smaller subset for human annotation allowed for subsequent automated filtration and augmentation to be explored on the remaining reports. By integrating human-verified labels with additional, augmented data, this approach maximized both data quality and diversity, enabling the model to better capture the complexity of PCC literature.

**Step 1 - Initial entity extraction using pre-trained NER models.** Two pre-trained NER models from JSL were applied in parallel to the full set of model development case reports to generate preliminary entity annotations. The first model, "ner_sdoh" [46], was trained to identify 47 entities spanning social and clinical factors, while the second model, "ner_covid_trials" [47], was trained to categorize 41 entities including attributes relevant to COVID-19. Based on the alignment of "ner_sdoh" with our proposed objective, "ner_sdoh" was utilized as the base model and annotations from "ner_covid_trials" replaced those from "ner_sdoh" for certain critical entities. Namely, initial explorations revealed a superior performance of "ner_covid_trials" in the age entity. Vaccination status, which was not included in "ner_sdoh," was also included due to its relevance to PCC. This substitution enhanced dataset relevance by ensuring that key entities were more accurately represented.

**Step 2 - Human annotation.** Following initial extraction, subset 1, comprising 99 case reports, was selected to undergo human annotation. This subset was curated to include adequate representation of a variety of entities of interest, with particular emphasis on keywords related to marginalized and at-risk populations to ensure diverse representation in the training data. The subset was divided equally among three human annotators tasked with reviewing and refining the model-predicted labels. Before annotation, the annotators reached a consensus on entity definitions and labeling criteria, which was essential for consistency and reduced inter-annotator variability. This preparatory step ensured uniformity in label interpretation, thus enhancing the reliability and precision of the final annotated dataset.

**Step 3 - Entity type refinement.** Following expert advice, a review of the WHO's SDOH framework [48], and under the guidance of three human annotators with health-related experience, we implemented an entity type refinement process. This was aimed at enhancing our model's classification accuracy while preserving critical sociodemographic dimensions relevant to our study. Importantly, this process entailed combining highly interrelated labels and eliminating entities outside of the research focus of this analysis. The entity label dimensions included, excluded and refined are summarized in Table 1.

The base models from JSL initially yielded over 45 distinct entities (resulting in 90 labels in CoNLL format), of which 14 (28 in CoNLL format) were identified as outside of the scope of this analysis. More details on the refinement process are outlined in supplementary materials

**Table 1. Entity label refinement process.**

| Entity Label Dimensions Excluded (Classified as 'O') from Pretrained Models | Entity Label Dimensions Included from Pretrained Models | Refined Entity Labels |
|---|---|---|
| "Community Safety"<br>"Date"<br>"Healthcare Institution"<br>"Legal Issues"<br>"Other SDoH Keywords"<br>"Population Group"<br>"Quality Of Life"<br>"Sexual Activity"<br>"Substance Duration"<br>"Substance Frequency"<br>"Substance Quantity"<br>"Transportation"<br>"Childhood Event"<br>"Environmental Condition" | "Access To Care"<br>"Age"<br>"Alcohol"<br>"Communicable Disease"<br>"Diet"<br>"Disability"<br>"Eating Disorder"<br>"Education"<br>"Employment"<br>"Exercise"<br>"Family Member"<br>"Financial Status"<br>"Food Insecurity"<br>"Gender"<br>"Geographic Entity"<br>"Housing"<br>"Hyperlipidemia"<br>"Hypertension"<br>"Income"<br>"Insurance Status"<br>"Language"<br>"Marital Status"<br>"Mental Health"<br>"Obesity"<br>"Other Disease"<br>"Race Ethnicity"<br>"Sexual Orientation"<br>"Smoking"<br>"Social Exclusion"<br>"Social Support"<br>"Spiritual Beliefs"<br>"Substance Use"<br>"Transportation"<br>"Violence Or Abuse"<br>"Vaccine"<br>"Admission Discharge" | "Access To Care"<br>"Age"<br>"Condition"<br>"Diet"<br>"Disability"<br>"Education"<br>"Employment"<br>"Exercise"<br>"Family Member"<br>"Gender"<br>"Geographic Entity"<br>"Housing"<br>"Income"<br>"Insurance Status"<br>"Language"<br>"Marital Status"<br>"Mental Health"<br>"Race Ethnicity"<br>"Severity"<br>"Sexual Orientation"<br>"Social Support"<br>"Spiritual Beliefs"<br>"Treatment"<br>"Vaccine"<br>"Violence Or Abuse"<br>"O" |

(sFigure 1). This refinement resulted in a set of 27 core entity types. For CoNLL format consistency, each label (except for 'O') was divided into 'B-' (beginning) and 'I-' (inside) sub-labels, resulting in a total of 53 distinct types for model training.

 **Step 4 - Filtration and data augmentation.** Addressing the class imbalance, particularly the over-representation of the non-entity ('O') class, was essential for improving entity extraction accuracy. To tackle this, a two-stage filtration and augmentation strategy was implemented on subsets 2 and 3, aimed at enhancing the representation of rare entities while mitigating class imbalance. For the first stage, entity types with a probability below 90% were first reassigned to the 'O' class for added robustness. Subsequently, the entity types were reduced and refined following the process outlined in step 3. In addition, to reduce misclassification noise from numeric values, any numeric labels not identified as 'Age' (either 'B-Age' or 'I-Age') were also converted to 'O'. To further address the prevalence of the no-entity ('O') class during training, sentences containing only 'O' class labels were removed from subset 2 alone, while subset 3, which is utilized for evaluation, was not subjected to this additional sentence-level downsampling technique.

The augmentation process was applied separately to subset 2, designated for training, and subset 3, designated for evaluation. For the training subset, three sentence templates were generated with GPT-4o [49] to minimize the presence of the 'O' class while retaining all entity types of interest. Both sentence templates and variations were manually reviewed for potential biases, and a manual review of a subset of the produced sentences ensured that there was not a disproportionate prevalence of a particular variation. These templates were tokenized, labeled, and stored in three base dataframes. From these structures, 3,000 sets of synthetic sentences were created by randomly combining rare (non-'O') entities, with 1,500 derived from a complete entity variation dictionary (composed of the set of rare entities in the corpus and variations generated by GPT-4o) and 1,500 derived from the variations generated by GPT-4o alone. Each synthetic sentence set was assigned a unique identifier and was randomly embedded within the case reports, seamlessly integrating with the dataset for model training. This augmented 402 case reports was then merged with the 99 human-reviewed case reports to form the final training dataset for the NER model training. To assess potential model overfitting and improve generalizability, the augmentation of subset 3 was designed with greater class sparsity and distinct sentence structures (i.e., different entity type orders) than those used in training. GPT-4 generated three distinct sentence templates, featuring higher 'O' class representation and varied label sequences. Importantly, gender was not included in all sentence structures given its greater presence pre-augmentation. As before, each sentence structure was tokenized, labeled, and structured into three base dataframes. Using these dataframes, 3,000 augmented sets of sentences were generated, with 1,500 sentences derived from a complete entity variation dictionary (composed of the set of rare entities in the corpus and variations generated by GPT-4o) and with 1,500 derived from the variations generated by GPT-4o alone. Each synthetic sentence was assigned a unique identifier and was embedded randomly into the evaluation case reports to evaluate model performance. The entity variations produced by GPT-4o and sample prompts utilized to produce the entity type variations and sentence structures are outlined in supplementary materials.

## NER model development

Building on the annotated corpus, we trained and evaluated several NER models to automatically extract SDOH-related attributes from case reports. This step allowed us to scale the extraction process across thousands of documents.

**NER models.** In this study, both transformer-based models and traditional deep learning architectures are benchmarked to assess their capabilities for the selected NER task, with a particular emphasis on handling imbalanced entity classes and generalizing to unseen textual structures. The primary models of interest were BERT-based, encoder-only transformer models, due to their encoder architecture, which is especially suited for NER tasks due to its transformer encoder-only architecture [50]. In particular, BERT-base-uncased [51,52] and DistilBERT-base-uncased [53] were selected due to their substantial pre-training on English language texts. In addition, the Biomed-NLP model, which utilizes the BERT architecture, was included in the model exploration due to its pre-training on a large body of biomedical texts [54]. We also include BioBERT and Biomedical NER (BERT-based) due to their vast pre-training on biomedical data [55,56]. As baselines, Bidirectional LSTM (BiLSTM), Recurrent Neural Network (RNN), and Gated Recurrent Unit (GRU) models are included due to their ability to process sequential dependencies [57–59].

For NER model building, to accommodate the 512-token limit per textual sample, each full string (case report text, plus augmentation in selected cases) was split into segments of

up to 512 tokens while ensuring the preservation of sentence boundaries, to minimize contextual loss. For the BERT-based models, 512-token sequences were prepared with CLS, SEP, and PAD tokens, along with a mask layer. Padded sequences were generated for the BiLSTM, RNN, and GRU models.

All BERT models were fine-tuned for 300 epochs with a learning rate of 1e-5. Other deep learning models were fully trained from random initialization for 10 epochs with a learning rate of 1e-9. All models employed a sparse cross-entropy loss function and were optimized using the Adam optimizer, with regularization techniques such as class weighting, sample weighting, and dropout layers to mitigate overfitting on majority classes. Importantly, a lower learning rate was implemented for the training of non-transformer models due to an observed propensity to overfit on the training data during early experiments. Conversely, to optimize the performance of transformer models, such that their performance would more closely assimilate that of non-transformer models on the test set, a higher number of epochs was implemented during training.

For model assessment, the macro F1 score is utilized as the primary performance metric. To account for the influence of the 'O' class performance in the macro F1 score, the macro F1 score excluding the 'O' class is also assessed. Other key performance metrics assessed include the macro One vs. One (OVO) Area Under the Receiver Operating Characteristic Curve (AUC) and the macro One vs. Rest (OVR) AUC scores. For the best-performing model, we also present class-specific F1 scores.

## NLI pipeline development

To complement NER and gain more detailed insights into the extracted entities and their corresponding types, a rule-based NLI pipeline was leveraged. Importantly, the entities were utilized as the strings to be assessed, and statements were crafted by entity type to assess entailment, contradiction and neutrality/non-applicability of the entities. These statements are outlined in the supplementary materials. Given the significant variation in meaning and scope among our entity types, the selected statements encompass both the presence and absence of attributes, as well as other meaningful binary distinctions in the data. To support this analysis, we generated lists of words indicating 'entailment' and 'contradiction' for each entity using prompts in GPT-4. We further enhanced these lists by augmenting them with synonyms from the NLTK library [60]. Using RegEx each extracted entity was matched against the generated lists, returning 'Entailment,' 'Contradiction,' or 'Not Applicable' as appropriate. Importantly, only tokens marked with 'Entailment' or 'Contradiction' are considered, given that those marked as 'Not Applicable' are prone to contain noise and other variations out of the scope of this pipeline. Manual validation on a subset of matches was performed to assess RegEx accuracy and refine patterns. The entailment and contradiction sets (produced by GPT-4o), and the corresponding prompt, are outlined in the supplementary materials.

**Statement selection.**   For entity types including 'Social_Support,' 'Substance,' 'Marital_Status,' 'Disability,' 'Housing,' 'Insurance_Status,' 'Violence_Or_Abuse,' 'Employment,' 'Vaccine,' 'Mental_Health,' and 'Access_To_Care,', the statements indicate the presence or absence of attributes.

For more complex labels representing multiple categories or continuums, the statements focus on insights related to sensitive or at-risk groups. or to highlight other relevant binary breaks in the data. For 'Education,' the statement was crafted to create a break between terms specifying a higher formal education level (high school or above) from those that do not. Importantly, having a general education development (GED) diploma has been associated with better health outcomes later in life [61]. For 'Race_Ethnicity,' the statement generates a

binary break between white/caucasian and non-white, which is significant given the mechanisms of structural racism and their effects on the health experience and outcome of non-white individuals [62]. We recognize that this binary classification is limited, as it may not capture intersectional nuances; however, we justify this choice based on the goals of the study to provide initial insights for further study. For 'Exercise', the statement is aligned with the documented effect of exercising regularly on the immune response, particularly with regard to COVID-19 [63]. For 'Geographic_Entity' and 'Language,' the statements were chosen in conjunction to reflect the effect of a language barrier on the health care experience, such that it can be observed if there is a mismatch between the language spoken in the country of origin/care location and the primary language [64]. Importantly, both statements are dichotomized as 'English or not' given that our selected texts are in English, assessing, therefore, if there is an English language dominance in the location and language of subjects in the case reports. For 'Severity,' the statement is crafted to identify if the observed symptoms are highly severe or not, which can be of interest to observed potential patterns between long COVID and symptom/condition severity.

In addition, for 'Diet,' the statement aims to capture the relationship between dietary restrictions and health, which has been associated with immune responses and other health conditions [65,66]. For 'Income,' the statement was crafted to reflect the effect of higher income level (or lack thereof) on access to health care [67]. Given that our corpus hails from distinct countries, with distinct thresholds for upper-middle income, we do not include numeric measures in the entailment and contradiction sets. For 'Family_Member,' the statement was crafted to capture the social support provided by progeny, which is an essential form of caretaking in multiple cultural settings, with an associated impact on parental health [68]. For 'Sexual_Orietation,' the statement was selected to align with the discourse of discrimination and barriers to care faced by non-heterosexual individuals [69]. For 'Gender,' the statement was selected to reflect the discussed positive associational relationship between female gender, sex on the development of PCC [15,16]. For 'Spiritual_Beliefs,' the statement was crafted to align with the historical privilege faced by those of catholic and christian belief systems [70], and the lack thereof faced by several other faith systems, which can be reflected in experiences of discrimination in health care settings [71]. Lastly, for 'Age,' 'Treatment,' 'Condition' and 'Severity' the statements were crafted to highlight at-risk groups including older adults and other inmuno-compromised groups, such as those with a highly invasive treatment and those with highly severe, rare, chronic or terminal conditions [72,73].

## 4 Results

### Distribution of entity types across case reports

The distribution of all 26 ground truth entity types (excluding the NuLL entity) across the case reports for the model training and generalization evaluation before and after augmentation is illustrated in Fig 2. Post-augmentation, the model training set and the generalization evaluation set contain 693,476 and 711,743 entities, respectively. The complete collection of case reports contains 1,405,219 entities. 'Condition', 'Gender', 'Access to care', 'Age', and 'Employment' are the most frequent entities in the 709 case reports leveraged for model training and evaluation. The greater presence of these entity labels reflect biases and oversimplification of SDOH entity dimensions in PCC case reports. As the least prevalent entities, label dimensions such as 'Spiritual_Beliefs' and 'Sexual_Orientation' saw the greatest effect in variational diversity and relative frequency post-augmentation (increasing from 0.0% to over 0.8% on the training and generalization evaluation sets).

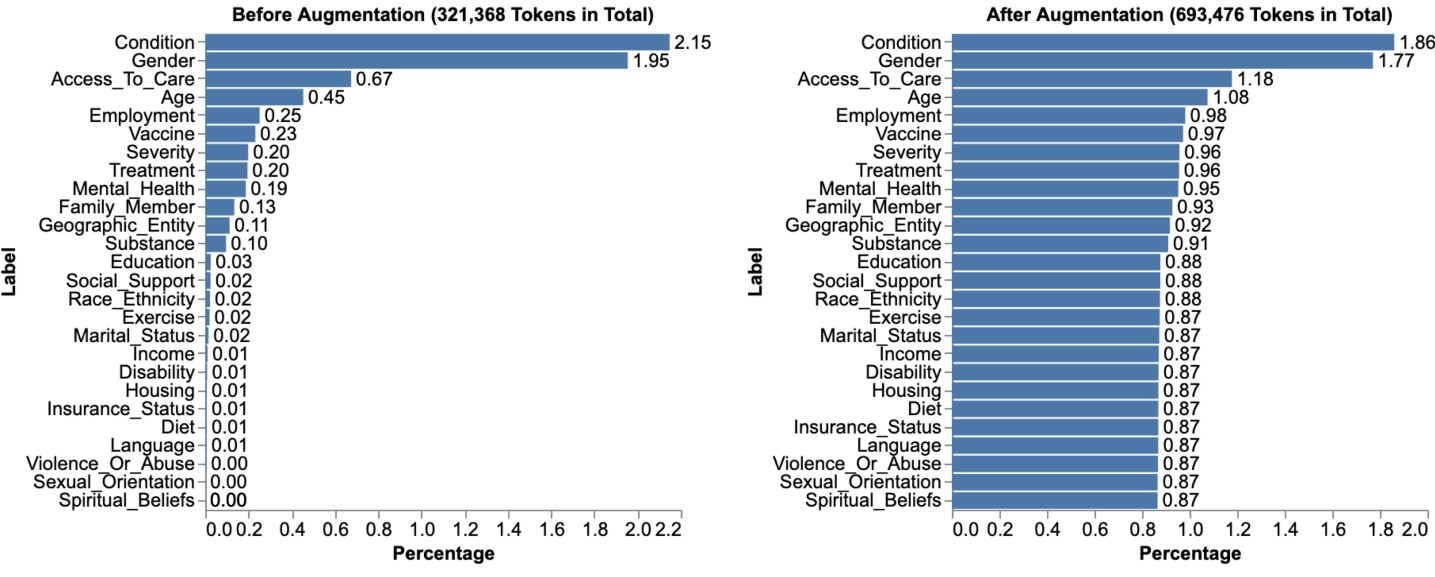

(a)

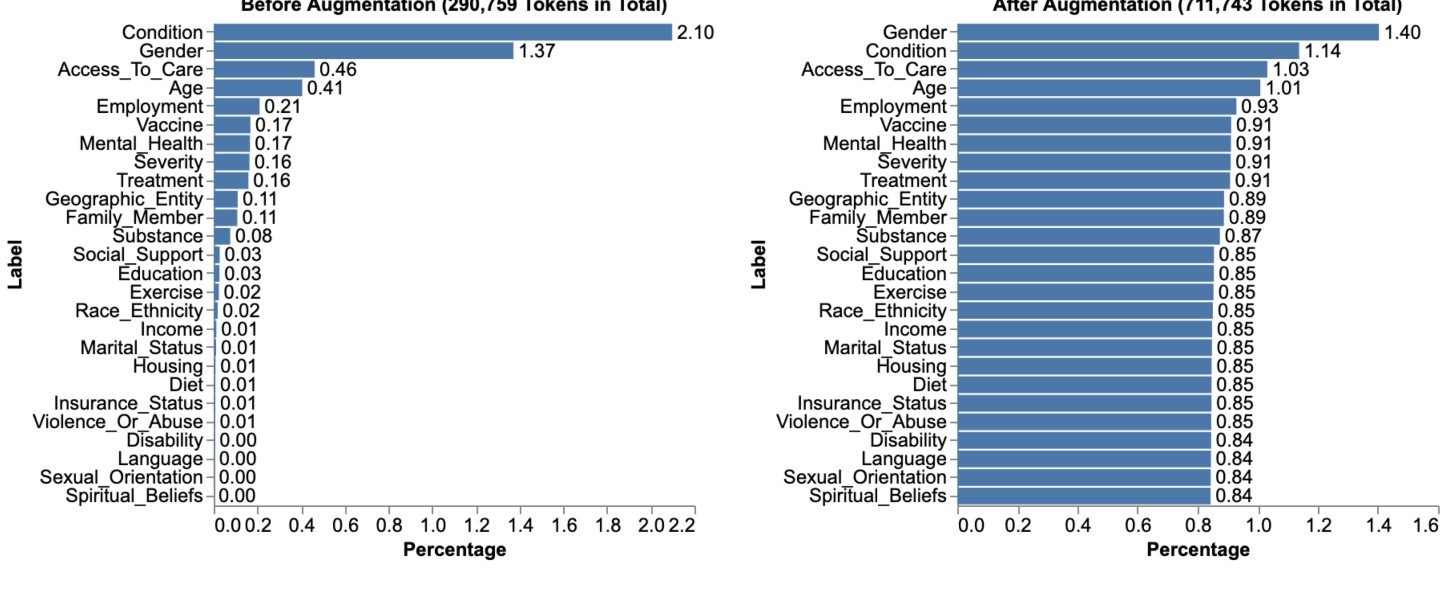

(b)

**Fig 2. Entity frequencies (percentage) for (a) model training and (b) evaluation case reports before and after augmentation.** The percentages are calculated out of the total number of tokens, which were 321,368 before augmentation and 693,476 after augmentation for model training and 290,759 before augmentation and 711,743 after augmentation for model evaluation.

## Comparative benchmarking analysis of NER models

To evaluate how effectively different NER models capture SDOH information from PCC case reports, we benchmarked BiLSTM, BERT, DistilBERT, BioBERT, BiomedNLP and BiomedicalNER. Their performance was compared across 27 entity types using both the optimization test set and an independent generalization evaluation set, as summarized in Table 2. As the primary metric of model assessment, macro F1 score, excluding the 'O' class, is utilized. Excluding the 'O' class reduces bias from overrepresented non-entity annotations, highlighting the model's ability to accurately classify diverse entity types. The macro-average results for these benchmarks are presented, demonstrating the comparative performance of each model in this task. The performance of additional RNN-based models is noted in the supplementary material. Overall, we show that BERT models are on average more generalizable than the RNN-based models, although RNN models outperform BERT for the optimization testing set.

In particular, the BERT-base-uncased model showed the strongest performance on the Generalization Evaluation Set. Notably, the model demonstrated strong performance in classes such as 'I-Race_Ethnicity' and 'I-Marital_Status', both achieving an F1-Score of 0.99, indicating high precision and recall (supplementary materials). Other well-predicted classes included 'I-Language' and 'I-Spiritual_Beliefs' (0.98), and 'B-Education' and 'I-Violence_Or_Abuse' (0.95). On the contrary, the least predicted classes had notably lower performance. 'B-Condition' and 'I-Condition' were the lowest, with F1-scores of 0.17 and 0.14, respectively, indicating difficulties in recognizing specific health conditions. 'B-Gender' and classes like 'B-Diet' and 'B-Insurance_Status' also underperformed, with scores below 0.40. Exploratory analysis of a subset of classifications in each class revealed that misclassifications stem from model limitations to various health conditions, and to ambiguous mentions and tokenization artifacts.

## Exploratory analysis of the entities extracted by the best-performing model

The distribution of extracted entities offers valuable insight into how social and behavioral factors are represented in PCC case reports. Using the best-performing model on the Generalization Evaluation Set, a fine-tuned BERT-base-uncased, we applied automated extraction across the full corpus (7,172) to quantify the presence and diversity of 1,369,863 extracted entities across 26 distinct entity type dimensions, excluding the 'O' entity (amounting to 2,097,047 entities alone). The distribution of entity counts per case report is illustrated in Fig 3a. We observed variability in entity richness across the corpus, ranging from 6 to 26 distinct non-'O' types. The majority of case reports contained 18 to 20 distinct entity types, with

Table 2. The optimization testing set and the generalization evaluation set. The optimization testing set consists of a randomly selected 20 percent subset of the training data (501 case reports and 3000 sets of synthetic sentences), while the generalization evaluation set is an entirely separate subset from the case report cohort, including 208 case reports and 3000 sets of synthetic sentences. OVO denotes One vs. One macro AUC scores. BERT is the Base Uncased model.

| Model | Optimization testing set | | Generalization evaluation set | |
|---|---|---|---|---|
| | Macro F1-score | Macro AUC OVO | Macro F1-score | Macro AUC OVO |
| BERT | 0.78 | 0.78 | **0.72** | **0.72** |
| DistilBERT | 0.79 | 0.78 | 0.63 | 0.62 |
| BiomedicalNER | 0.64 | 0.99 | 0.52 | 0.98 |
| BioBERT | 0.48 | 0.98 | 0.36 | 0.94 |
| BiomedNLP | 0.58 | 0.58 | 0.28 | 0.27 |
| BiLSTM | **0.95** | **0.95** | 0.10 | 0.09 |

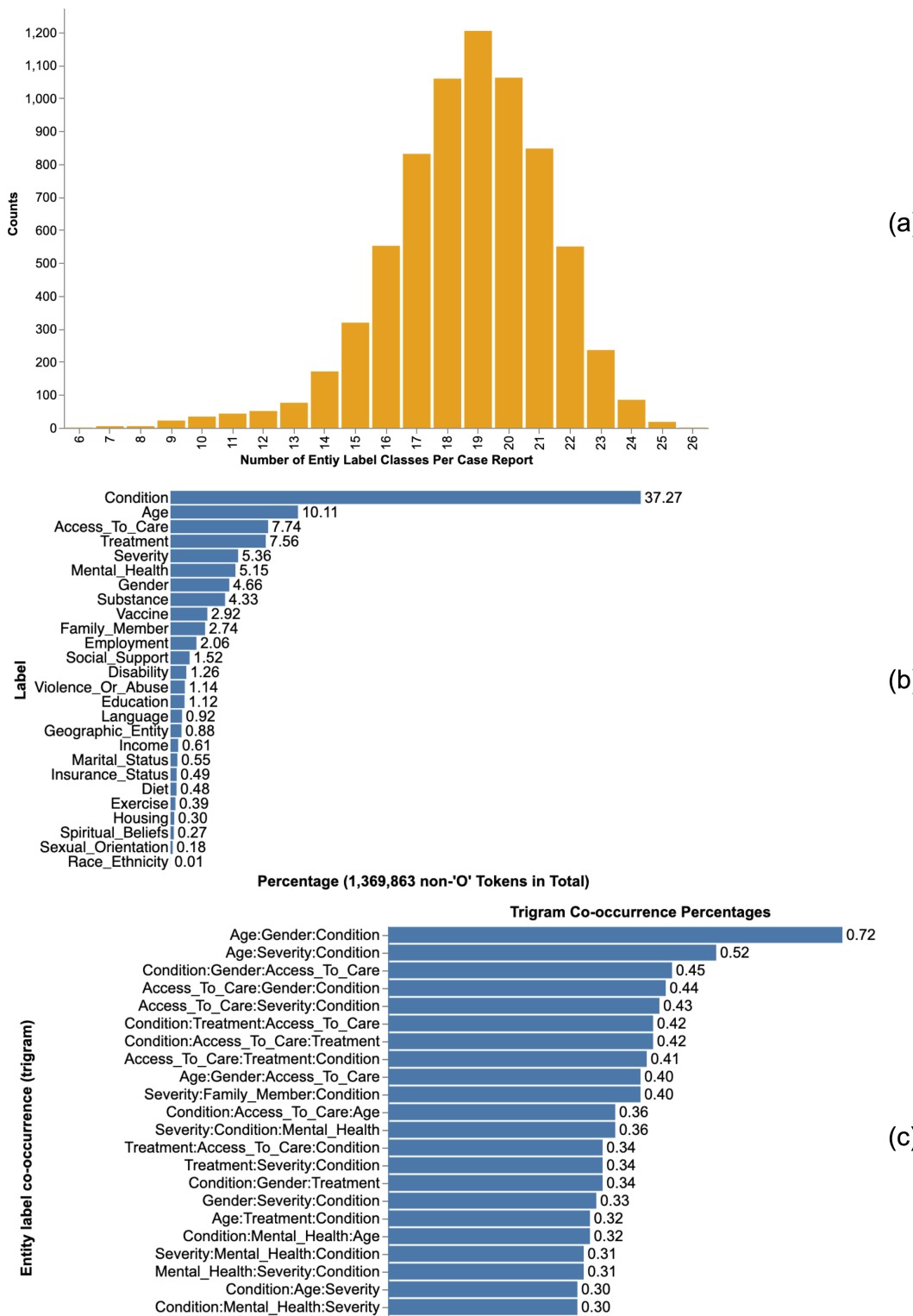

**Fig 3. Entity extraction and distribution analysis from 7,172 case report sections processed by the fine-tuned BERT Base Uncased model.** (a) Distribution of entity counts per case report. (b) Frequency of non-'O' entity labels, out of all non-'O' entities. (c) Trigram frequency visualization for the top 25 most frequent entity types, where the order of occurrence is considered.

19 being the most common, observed in 1,204 case reports. A small subset of reports (18) included 25 distinct entity types, while only one case report encompassed all 26 non-'O' entity types.

The frequency of non-'O' entity types, out of all non-'O' entities is depicted in Fig 3b. The most frequently mentioned categories include biological components labelled as 'Condition' (37.27%), 'Age' (10.11%), 'Access_To_Care' (7.74%), 'Treatment' (7.56%), 'Severity' (5.36%), 'Mental_Health' (5.15%), 'Gender' (4.66%), and 'Substance' (4.33%). Conversely, entities such as 'sexual orientation' (0.18%), 'spiritual beliefs' (0.27%), 'Race_Ethnicity' (0.01%), and 'housing status' (0.30%) are among the least represented, aligning with their sensitive nature.

In addition to analyzing the individual occurrence of the top 25 most common entity types, we examined their co-occurrence patterns by visualizing ordered three-entity co-occurring sequences (trigrams) in Fig 3c, where the order of occurrence is considered. Among the most frequent entity type trigrams, 'Age, Gender, Condition' stands out with a frequency of 0.72%, followed by 'Age, Severity, Condition' at 0.52% and 'Condition:Gender:Access_To_Care' at 0.45%. Other notable trigrams include combinations of 'Access_To_Care', 'Gender', 'Condition', 'Severity', 'Treatment', 'Age', 'Family_Member', 'Mental_Health' from 0.44% to 0.28%.

### Entailment and contradiction patterns across key attributes

Beyond occurrence and trigram analysis, understanding the relationships between entities, such as entailment and contradiction is essential for capturing deeper contextual meaning. All extracted entities were grouped by their respective entity types and matched with the corresponding statements allowing for a clear comparison of how each entity aligns with or contradicts the defined relationships. The analysis of entailment and contradiction across various entity types reveals several key trends (Fig 4a). Notably, *"Experienced violence or abuse"* and *"Has medical insurance"* are predominantly entailed, at 82.4% and 80.3%. Several attributes, including *"Utilizes psychoactive substances,"* *"Has social support,"* *"Exercises regularly,"* *"Is employed,"* and *"Has access to care,"* display moderate levels of entailment while also showing notable contradictions. In contrast, attributes such as *"Is a senior adult,"* *"Is female-identifying,"* *"Is married,"* *"Is heterosexual,"* and *"Has a terminal, rare, or chronic condition"* exhibit high levels of contradiction, with contradiction rates reaching as high as 97.5%, 98.5%, 70.8%, 81.8%, and 88.5%. Other attributes, such as *"Is homeless,"* *"Has high school education,"* and *"Is white/Caucasian,"* show more variability in their levels of entailment and contradiction. To further evaluate whether these patterns persist in the full-text context, we removed prior matching restrictions on the extracted entities (Fig 4b). This comparison revealed consistent trends across most entity types, regardless of the applied restrictions.

### 5 Discussion

Our study demonstrates the potential of LLMs, particularly BERT-based models, in addressing SDOH-specific NER tasks. The use of NLP to gain insights into health-related problems has been an area of research since the 1950s [74], but its large-scale application gained significant momentum during the COVID-19 pandemic. This shift was exemplified by initiatives such as the CORD-19 competition [75,76], which coincided with the emergence of transformer-based models [77] and significant advancements in NLP methodologies and applications [78]. These developments marked a transformative era in NLP, enabling unprecedented breakthroughs in understanding and analyzing health data [79,80]. By proposing a generalizable framework for extracting these attributes, we demonstrate the viability of leveraging LLMs for this purpose. Notably, our findings indicate that BERT-based encoder-only

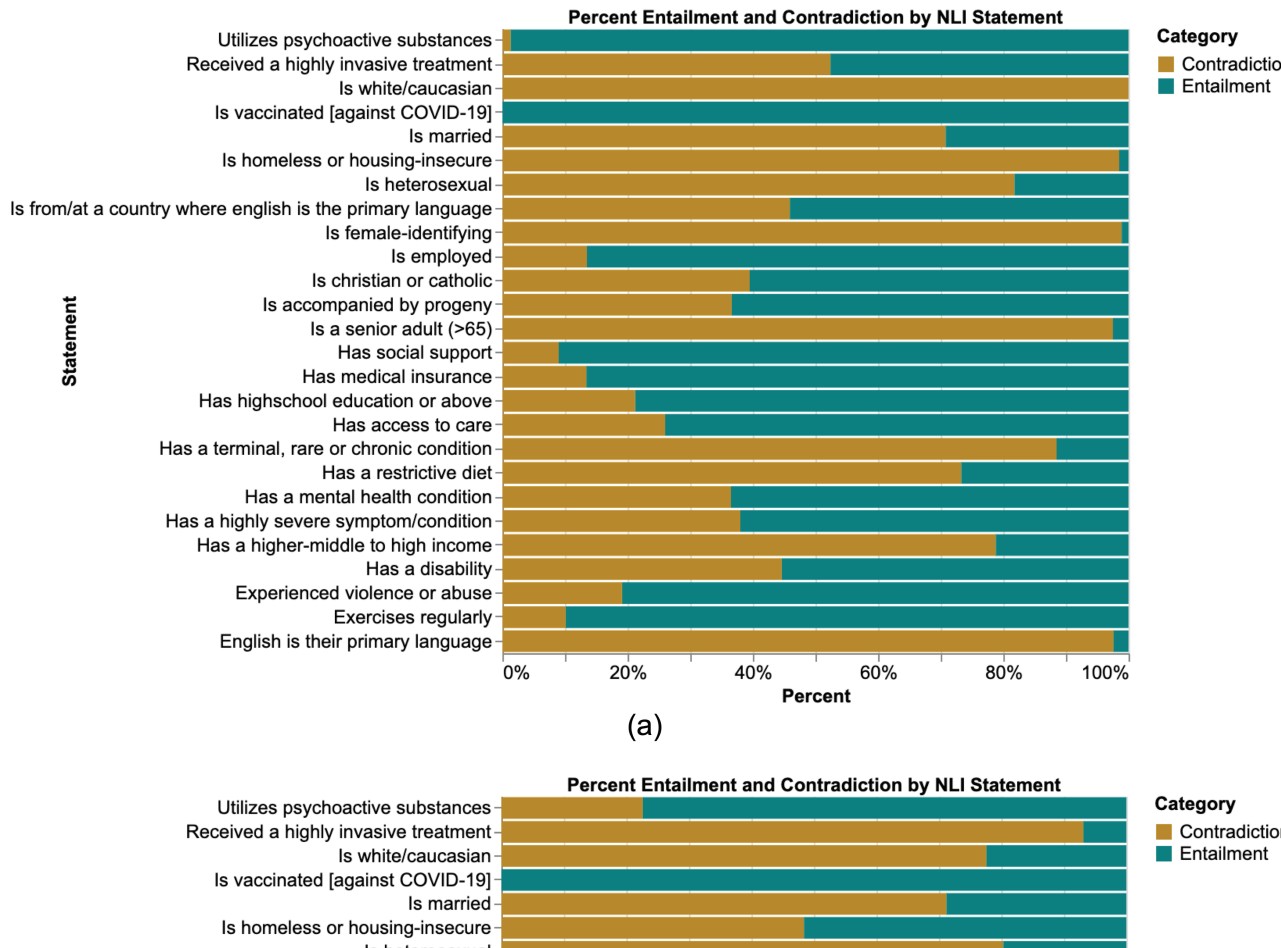

**Fig 4. (a) Percent entailment and contradiction by NLI statement with RegEx matching within model-predicted label dimensions.** (b) Percent entailment and contradiction by NLI statement, with RegEx matching on the full dataset.

models achieved superior classification performance on the generalization set, underscoring their efficacy in handling complex, domain-specific NER tasks. Previous studies have demonstrated the superior generalization capabilities of BERT models compared to both traditional deep learning and also decoder based transformer models in NER tasks, largely due to their self-attention mechanisms [81,82]. Our findings align with these benchmarks, providing further evidence of BERT's robustness in handling unseen sentence structures and addressing class sparsity [51,53]. In contrast, the comparatively lower performance of the BiomedNLP model is attributed to its pre-training on domain-specific biomedical texts, which limits its flexibility and adaptability [54].

The exploratory analysis revealed significant gaps in sociodemographic representation within PCC case reports, with attributes like race, spiritual beliefs, and housing status appearing infrequently. In contrast, age, gender, mental health, and access to care were more frequently discussed. Additionally, the co-occurrence patterns of biomedical and sociodemographic factors often followed a structured narrative, with age and gender commonly contextualizing conditions and symptoms. While case report sections typically contained 18 to 20 non-'O' distinct entities dimensions, it was rare for a report to encompass all dimensions; only one of 7,172 reports included predictions for all 26 identified labels. These findings align with prior research highlighting the scarcity of robust sociodemographic representation in clinical PCC datasets [21–24]. Our analysis confirms that academic PCC-related case reports similarly lack diversity across multiple attributes, further underscoring the need for improved documentation and reporting practices to enable more equitable and comprehensive analyses in PCC research. Lastly, an analysis of the most frequent label trigrams reveals that '*Condition*' commonly appears in the final position, preceded by entities such as '*Age*', '*Gender*', and '*Severity*'. This pattern suggests two key findings: first, academic case reports often adopt a structured approach to representing the patient experience; second, this representation is frequently constrained to a limited subset of sociodemographic factors.

Our key findings highlighted that mentions of attributes such as 'having access to care', 'being vaccinated against COVID-19', 'exercising regularly', 'having medical insurance', 'social support', and 'experiencing violence or abuse' were predominantly in agreement, reflecting positive representation. In contrast, mentions of being 'female-identifying', 'white/Caucasian', 'heterosexual', 'senior', or 'having a terminal, rare, or chronic condition' were most frequently classified as contradictory to the statement, suggesting potential underreporting or biases. Some of these findings highlight discrepancies with prior research, particularly concerning the underrepresentation or absence of mentions related to being female-identifying, unvaccinated for COVID-19, or a senior adult. Previous studies have identified these groups as potential at-risk populations in the COVID-19 context, with discrimination being significantly associated with prolonged COVID-19 symptoms [15,16,18,72,83]. Research has also linked female gender and sex to a higher prevalence of PCC [15,16] and highlighted socioeconomic status as a significant factor in PCC development [17]. However, these insights should not be interpreted as evidence against previous findings. Instead, they may point to underlying biases in academic research and structural barriers that deter these groups from participating in clinical studies. This aligns with earlier work highlighting selection bias and the underrepresentation of older adults and women in clinical research [84,85]. Additionally, individuals unvaccinated for COVID-19 often face societal discrimination, which may further impact their access to care and inclusion in research [86]. These patterns underscore the need to address systemic biases in data collection and study design to ensure more equitable representation, such as standardizing sociodemographic reporting protocols and promoting the inclusion of underrepresented groups in clinical research.

## Limitations and future work

While our NLP approach effectively identifies many sociodemographic determinants in a PCC-related corpus, its performance in predicting medical symptoms and conditions remains limited. This challenge arises from the vast diversity of diseases and symptoms, which would require pretraining on a large, domain-specific medical text corpus. Our experiments with BiomedNLP, BioBERT and Biomedical NER demonstrated low performance in the NER task, however, highlighting the need for better-tailored models. The analysis is inherently influenced by corpus-related biases, including a focus on English-language texts and an over-representation of individuals with access to care, in addition to biases associated with GPT. Additionally, while we attempted to comprehensively categorize entailment and contradiction tokens using synonyms, some relevant terms may have been overlooked. Importantly, the distributions highlighted by our analysis should not be interpreted as robust associations but rather as exploratory insights.

To address these limitations, future efforts should focus on enhancing the documentation of underrepresented attributes, such as race, housing status, and sexual orientation, to support equitable AI-driven analyses in PCC research. Future work could involve pretraining a BERT or other transformer models on a larger PCC-specific corpus to improve performance, though this would require significant computational resources. Evaluating additional biomedical models beyond BiomedNLP, BioBERT and Biomedical NER may also enhance NER outcomes. To better understand interactions between entities, future research could develop an ML-driven relation extraction (RE) pipeline to explore how SDOH factors interrelate. Extending this analysis to other patient populations would provide a broader understanding of sociodemographic mentions across diverse health contexts. Additionally, the insights into PCC-entity representation could inform policy discussions on diversity and inclusion in academic research. Adapting the pipeline to other health domains could also offer a more comprehensive view of sociodemographic reporting in case reports. Finally, given the evolving definition of social determinants of health, all future work must entail an assessment of the relevance and comprehensiveness of the included entity label dimensions.

**Real-world applicability of the proposed dataset.** The novel dataset presented in this study provides labeled data for 27 entity label dimensions in over 7,000 academic case report sections related to Post COVID Condition. Real-world utilization of this dataset includes leveraging label frequencies as support diversity and inclusion efforts in PCC research, particularly those relating to gender-based barriers and disparities. In addition, the dataset can be utilized for training and further fine-tuning of named-entity-recognition models in the PCC context. Lastly, the development of visual stories based on the presented data, coupled with research developments on the effect of social determinants on PCC development, would help engage stakeholders on the complex dynamics of SDOH in the PCC context.

## 6 Conclusion

This analysis benchmarks traditional deep learning models and encoder-only transformer models for Named Entity Recognition of sociodemographic entities in academic case reports related to Post COVID-19 Condition. Importantly, it not only provides a model comparison assessment, but provides a comprehensive pipeline equipped with various annotation, augmentation, cleaning and regularization techniques, culminating in a timely model that can be utilized by researchers in the field. Notably, encoder-only transformer models were found to outperform traditional deep learning models on unseen data with distinct sentence structures, and greater class sparsity. Furthermore, the best performing model on the validation data (BERT base uncased) was applied to all 7,172 extracted case reports. Insights were

extracted from the annotated data through exploratory techniques and through a rule-based natural language inference (NLI) pipeline leveraging variations generated by GPT-4o for RegEx matching. Ultimately, we conclude that there is scarcity in multiple sociodemographic factors in PCC-related academic case reports, and an imbalanced representation of mentions aligned with groups of interest within dimensions such as gender, insurance status, and age.

## Supporting information

**S1 Appendix A. Corpus construction annotation details.**

**S1 A.1 Additional explorations on the corpus with GPT.**

**S1 A.2 Entity label refinement.**

**S1 A.3 Sample prompt to GPT-4o produce variations per entity type.**

**S1 A.4 Dictionary of variations per entity type (Produced by GPT-4o).**

**S1 A.5 Sample prompt to generate sentence structures for augmentation of the development set.**

**S1 A.6 Sample prompt to generate sentence structures for augmentation of the generalization set.**

**S1 Appendix B. SDOH extraction and analysis pipeline details.**
In this section we provide additional information on the NER and NLI pipelines.

**S1 B.1 Sample prompts to GPT-4o for entailment and contradiction set generation.**

**S1 B.2 NLI entailment and contradiction sets by entity type (generated by GPT-4o).**

**S1 B.3 STable 1. Natural Language Inference (NLI) statements by entity type dimension. The statements were crafted to represent meaningful binary distinctions in the data.**

**S1 B.5 STable 3. Performance of RNN and GRU models.**

**S1 B.6 Fined-Tuned BERT-Base-Uncased Model Configuration.**

## Author contributions

**Conceptualization:** Juan Andres Medina Florez, Shaina Raza, Elham Dolatabadi.

**Data curation:** Juan Andres Medina Florez, Shaina Raza, Rashida Lynn Ansell, Brendan T. Smith.

**Formal analysis:** Juan Andres Medina Florez.

**Funding acquisition:** Juan Andres Medina Florez, Elham Dolatabadi.

**Investigation:** Juan Andres Medina Florez.

**Methodology:** Juan Andres Medina Florez, Shaina Raza, Elham Dolatabadi.

**Project administration:** Juan Andres Medina Florez, Elham Dolatabadi.

**Software:** Juan Andres Medina Florez, Shaina Raza.

**Supervision:** Zahra Shakeri, Brendan T. Smith, Elham Dolatabadi.

**Validation:** Juan Andres Medina Florez, Shaina Raza, Elham Dolatabadi.

**Visualization:** Juan Andres Medina Florez.

**Writing – original draft:** Juan Andres Medina Florez.

**Writing – review & editing:** Juan Andres Medina Florez, Shaina Raza, Zahra Shakeri, Brendan T. Smith, Elham Dolatabadi.

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
