## [Decision Letter · Decision Letter 0]

17 Mar 2025

PONE-D-25-08167Academic case reports lack diversity: Assessing the presence and diversity of sociodemographic and behavioral factors related to Post COVID-19 ConditionPLOS ONE

Dear Dr. Dolatabadi,

Thank you for submitting your manuscript to PLOS ONE. After careful consideration, we feel that it has merit but does not fully meet PLOS ONE’s publication criteria as it currently stands. Therefore, we invite you to submit a revised version of the manuscript that addresses the points raised during the review process.

 As you can see with this email, the reviewers are commenting on the need of additional comparison experiments and formatting issues. The citation is not compulsory if not related. If the additional models are not  possible please discuss that in you response.

A rebuttal letter that responds to each point raised by the academic editor and reviewer(s). You should upload this letter as a separate file labeled 'Response to Reviewers'.A marked-up copy of your manuscript that highlights changes made to the original version. You should upload this as a separate file labeled 'Revised Manuscript with Track Changes'. (**Highlighted in yellow with figures and tables inserted**)An unmarked version of your revised paper without tracked changes. You should upload this as a separate file labeled 'Manuscript'.

We look forward to receiving your revised manuscript.

Kind regards,

Issa Atoum

Academic Editor

PLOS ONE

Journal Requirements:

2**. **Thank you for stating the following financial disclosure:

“Resources used in preparing this research were provided, in part, by the Province of Ontario, the Government of Canada through CIFAR, and companies sponsoring the Vector Institute www.vectorinstitute.ai/partnerships/. This publication was supported by the Canadian Institutes of Health Research (CIHR), Funding Reference Number 192124.

The CAN-TAP-TALENT is funded by the Canadian Institutes of Health Research (CIHR) – FRN 184898. The authors wish to acknowledge the CAN-TAP-TALENT for its role in supporting the completion of this CAN-TAP-TALENT Research Project.”

“Resources used in preparing this research were provided, in part, by the Province of Ontario, the Government of Canada through CIFAR, and companies sponsoring the Vector Institute www.vectorinstitute.ai/partnerships/. This publication was supported by the Canadian Institutes of Health Research (CIHR), Funding Reference Number 192124. The CAN-TAP-TALENT is funded by the Canadian Institutes of Health Research (CIHR) – FRN 184898. The authors wish to acknowledge the CAN-TAP-TALENT for its role in supporting the completion of this CAN-TAP-TALENT Research Project.”

“Resources used in preparing this research were provided, in part, by the Province of Ontario, the Government of Canada through CIFAR, and companies sponsoring the Vector Institute www.vectorinstitute.ai/partnerships/. This publication was supported by the Canadian Institutes of Health Research (CIHR), Funding Reference Number 192124.

The CAN-TAP-TALENT is funded by the Canadian Institutes of Health Research (CIHR) – FRN 184898. The authors wish to acknowledge the CAN-TAP-TALENT for its role in supporting the completion of this CAN-TAP-TALENT Research Project.”

4. Please amend the manuscript submission data (via Edit Submission) to include author Dr. Rashida Lynn.

5. Please amend your authorship list in your manuscript file to include author Dr. Shida Ansell.

Reviewers' comments:

Reviewer's Responses to Questions

**Comments to the Author**

1. Is the manuscript technically sound, and do the data support the conclusions?

Reviewer #1: Yes

Reviewer #2: Yes

2. Has the statistical analysis been performed appropriately and rigorously? 

Reviewer #1: Yes

Reviewer #2: Yes

3. Have the authors made all data underlying the findings in their manuscript fully available?

Reviewer #1: Yes

Reviewer #2: Yes

4. Is the manuscript presented in an intelligible fashion and written in standard English?

Reviewer #1: Yes

Reviewer #2: Yes

5. Review Comments to the Author

**Reviewer #1: **

The paper offers a significant contribution by introducing a new dataset and presenting intriguing results and findings. However, I firmly believe that addressing the following comments will be essential for the paper to be published:

1. The introduction lacks a clear presentation of motivation, failing to address the study's impact, research findings, and potential benefits.

2. It is advisable to include an Ethics Statement or a dedicated ethics section within the study.

3.The abstract requires enhancement by incorporating significant findings and insights derived from the conducted experiments.

4. Following the contributions section, it is recommended to include a section outlining the paper's structure, such as "Section 2 covers related work, section 3 details methodology."

5. The related work section is inadequate, merely stating that no research has been done on sociodemographic entities in COVID-19 contexts. The authors should consider discussing sociodemographic entities in other applications.

6. After presenting the related work section, it is crucial to highlight the uniqueness of the study by explaining how it differs from previous research.

7. The authors have not utilized models fine-tuned on biomedical data, such as BioBERT or biomedical longformer. A comprehensive analysis in this domain should include comparisons with at least seven or more models.

8. The inclusion of a section or subsection discussing the real-world applicability of the proposed dataset would enhance the paper's value.

9. The font size in Figure 4 can be increased to ensure values are visible.

10. While the authors have introduced the dataset, it is recommended to make the code publicly available for reproducibility. Additionally, more hyper-parameters should be presented beyond just the learning rate and number of epochs.

**Reviewer #2: **

Academic case reports lack diversity: Assessing the presence and diversity of

sociodemographic and behavioral factors related to Post COVID-19 Condition

The paper presents a very interesting and novel work. If authors could address the following comments it will significantly improve the paper to publishing state:

1) Please check with the flow of the paper. Section to section the flow should be maintained.

2) The introduction is very short. The introduction should present motivation, problem and solution. Also it should 3) present how you arrived at that particular solution given the problem.

4) The related work is very small. The authors have to present the related work so that the researchers 5) understand what others have contributed and helps the researchers to understand the uniqueness of the paper.

6) It is recommended to add hyperparameters so that the researchers in the community can reproduce the work or the paper.

7) Check with the font sizes of the figures in the paper. The figures should be readable and easy to understand.

8) It is recommended to implement more algorithms as baselines for better comparison this helps the researchers to analyse how well the proposed approach performs better.

9) The authors have to improve the abstract as well. It does not provide the results and contrituions in the abstract.

10) The data have been augmented with the LLMs. It would be better if authors provide the results with LLMs.

6. PLOS authors have the option to publish the peer review history of their article (what does this mean?). If published, this will include your full peer review and any attached files.

Reviewer #1: No

Reviewer #2: **Yes**

---

## [Author Response · Author response to Decision Letter 1]

2 Jun 2025

Subject: Response to PLOS ONE Revision (PONE-D-25-08167)

Dear Issa Atoum

Thank you sincerely for your quality check of our submission to PLOS ONE, entitled “Academic case reports lack diversity: Assessing the presence and diversity of sociodemographic and behavioral factors related to Post COVID-19 Condition”. We greatly appreciate the time and effort you have devoted to improving our manuscript.

Journal Requirements:

“Resources used in preparing this research were provided, in part, by the Province of Ontario, the Government of Canada through CIFAR, and companies sponsoring the Vector Institute www.vectorinstitute.ai/partnerships/. This publication was supported by the Canadian Institutes of Health Research (CIHR), Funding Reference Number 192124.

The CAN-TAP-TALENT is funded by the Canadian Institutes of Health Research (CIHR) – FRN 184898. The authors wish to acknowledge the CAN-TAP-TALENT for its role in supporting the completion of this CAN-TAP-TALENT Research Project.”

Thank you for pointing this out. We have revised our cover letter to include the following statement, as requested:

“Resources used in preparing this research were provided, in part, by the Province of Ontario, the Government of Canada through CIFAR, and companies sponsoring the Vector Institute www.vectorinstitute.ai/partnerships/. This publication was supported by the Canadian Institutes of Health Research (CIHR), Funding Reference Number 192124. The CAN-TAP-TALENT is funded by the Canadian Institutes of Health Research (CIHR) – FRN 184898. The authors wish to acknowledge the CAN-TAP-TALENT for its role in supporting the completion of this CAN-TAP-TALENT Research Project.”

“Resources used in preparing this research were provided, in part, by the Province of Ontario, the Government of Canada through CIFAR, and companies sponsoring the Vector Institute www.vectorinstitute.ai/partnerships/. This publication was supported by the Canadian Institutes of Health Research (CIHR), Funding Reference Number 192124.

The CAN-TAP-TALENT is funded by the Canadian Institutes of Health Research (CIHR) – FRN 184898. The authors wish to acknowledge the CAN-TAP-TALENT for its role in supporting the completion of this CAN-TAP-TALENT Research Project.”

Thank you for your guidance. As requested, we have removed all funding-related text from the Acknowledgments section of the manuscript. The following statement has been added:

4. Please amend the manuscript submission data (via Edit Submission) to include author Dr. Rashida Lynn. 5. Please amend your authorship list in your manuscript file to include author Dr. Shida Ansell.

Thank you for bringing this to our attention. We would like to clarify that Dr. Rashida Lynn and Dr. Shida Ansell refer to the same individual. Her full name is Rashida Lynn Ansell, and we have now updated the manuscript and the submission metadata to reflect her correct full name consistently throughout.

Thank you for your comment. We have revised the citation of the paper by Obadinma et al. (2025), as it has been published in npj digital medicine.

Reviewer #1:

The paper offers a significant contribution by introducing a new dataset and presenting intriguing results and findings. However, I firmly believe that addressing the following comments will be essential for the paper to be published:

Thank you for your insightful comments. In response, we have revised the Introduction to include a clear problem statement, articulate the study’s motivation, and present our proposed solution. We have also added content highlighting the study's impact, key research findings, and potential benefits to ensure a more comprehensive and compelling introduction.

2. It is advisable to include an Ethics Statement or a dedicated ethics section within the study.

Thank you for the helpful suggestion. We have now included an Ethics Statement in the manuscript. The added text is as follows:

This study exclusively utilized de-identified, publicly available case reports from the LitCOVID. (National Institute of Health. https://www.ncbi.nlm.nih.gov/research/coronavirus). The dataset is made available under the Open Database License (ODbL). As no human subjects were directly involved and no identifiable private information was accessed, this research did not require institutional ethics approval.

3. The abstract requires enhancement by incorporating significant findings and insights derived from the conducted experiments.

Thank you for the suggestion. We have revised the abstract to incorporate key findings and insights derived from our experiments, providing a clearer summary of the study’s contributions and results. For ease of review, the new text has been highlighted in yellow.

4. Following the contributions section, it is recommended to include a section outlining the paper's structure, such as "Section 2 covers related work, section 3 details methodology."

Thank you for the recommendation. We have added a brief outline of the paper’s structure following the Contributions section to guide the reader through the organization of the manuscript. For ease of review, the new text has been highlighted in yellow.

5. The related work section is inadequate, merely stating that no research has been done on sociodemographic entities in COVID-19 contexts. The authors should consider discussing sociodemographic entities in other applications.

Thank you for your comment. We have expanded the related work section to discuss the extraction of sociodemographic entities in other contexts.

6. After presenting the related work section, it is crucial to highlight the uniqueness of the study by explaining how it differs from previous research.

Thank you for your recommendation. We added a statement at the end of the related work section highlighting the uniqueness of the study. This is also complemented by the contributions outlined in the introduction.

7. The authors have not utilized models fine-tuned on biomedical data, such as BioBERT or biomedical longformer. A comprehensive analysis in this domain should include comparisons with at least seven or more models.

Thank you for your recommendation. We have added two additional models to the benchmark analysis, BioBERT and BiomedicalNER. In total the assessment comprises 8 models including BERT-Base-Uncased, DistilBERT-Base-Uncased, BioBERT, BiomedicalNER, BiomedNLP, BiLSTM, RNN and GRU. Note that performance of the RNN and GRU models is outlined in the supplementary material.

8. The inclusion of a section or subsection discussing the real-world applicability of the proposed dataset would enhance the paper's value.

Thank you for your recommendation. We have added a subsection within the “Limitations and Future Work” section highlighting the real-world applicability of the proposed dataset.

9. The font size in Figure 4 can be increased to ensure values are visible.

Thank you for your recommendation. We have increased the font size to increase visibility.

10. While the authors have introduced the dataset, it is recommended to make the code publicly available for reproducibility. Additionally, more hyper-parameters should be presented beyond just the learning rate and number of epochs.

Thank you for your comment. We have included the model configuration in the supplementary material. We will also make the code publicly available soon.

Reviewer #2:

Academic case reports lack diversity: Assessing the presence and diversity of sociodemographic and behavioral factors related to Post COVID-19 Condition. The paper presents a very interesting and novel work. If authors could address the following comments it will significantly improve the paper to publishing state:

1) Please check with the flow of the paper. Section to section the flow should be maintained.

Thank you for the feedback. To improve the overall flow of the manuscript, we have added introductory text at the beginning of each subsection in Sections 3 (Methods) and 4 (Results). These additions are intended to enhance clarity and ensure a smooth progression between sections. For ease of review, the new text has been highlighted in yellow.

2) The introduction is very short. The introduction should present motivation, problem and solution. Also it should 3) present how you arrived at that particular solution given the problem.

Thank you for your insightful comments. In response, we have revised the Introduction to include a clear problem statement, articulate the study’s motivation, and present our proposed solution. We have also added content highlighting the study's impact, key research findings, and potential benefits to ensure a more comprehensive and compelling introduction.

4) The related work is very small. The authors have to present the related work so that the researchers 5) understand what others have contributed and helps the researchers to understand the uniqueness of the paper.

Thank you for the suggestion. We have expanded the Related Work section to provide a clearer overview of existing contributions in the field and to better highlight the novelty and significance of our study.

6) It is recommended to add hyperparameters so that the researchers in the community can reproduce the work or the paper.

Thank you for your comment. We have added the model configuration to the supplementary material for reproducibility.

7) Check with the font sizes of the figures in the paper. The figures should be readable and easy to understand.

Thank you for your suggestion. We have increased the font sizes of the figures to increase readability.

8) It is recommended to implement more algorithms as baselines for better comparison this helps the researchers to analyse how well the proposed approach performs better.

Thank you for your suggestion. We have included two additional models (BioBERT and Biomedical NER) to the benchmark analysis. Also note that the performance of other baselines, namely, RNN and GRU, are outlined in the supplementary material.

9) The authors have to improve the abstract as well. It does not provide the results and contributions in the abstract.

Thank you for your comment. We have adjusted the subtract to include missing results and to highlight the contribution of the paper in facilitating the identification of diversity gaps in PCC research.

10) The data have been augmented with the LLMs. It would be better if authors provide the results with LLMs.

Thank you for your comment. The decision to augment the data with LLMs stems from a need to include data that contains multiple variations of each entity type in both the training and validation sets. Doing so is imperative to ensure that the model can adequately identify diverse variations, particularly in texts with vast scarcity in SDOH attributes (as is our case).

Editor Letter

PONE-D-25-08167

Academic case reports lack diversity: Assessing the presence and diversity of sociodemographic and behavioral factors related to Post COVID-19 Condition

PLOS ONE

Dear Dr. Dolatabadi,

Thank you for submitting your manuscript to PLOS ONE. After careful consideration, we feel that it has merit but does not fully meet PLOS ONE’s publication criteria as it currently stands. Therefore, we invite you to submit a revised version of the manuscript that addresses the points raised during the review process.

As you can see with this email, the reviewers are commenting on the need of additional comparison experiments and formatting issues. The citation is not compulsory if not related. If the additional models are not possible please discuss that in your response.

A marked-up copy of your manuscript that highlights changes made to the original version. You should upload this as a separate file labeled 'Revised Manuscript with Track Changes'. (Highlighted in yellow with figures and tables inserted)

We look forward to receiving your revised manuscript.

Kind regards,

Issa Atoum

Academic Editor

PLOS ONE

Login Page: https://www.editorialmanager.com/pone/default.aspx

---

## [Editor Report · Decision Letter 1]

4 Jun 2025

Academic case reports lack diversity: Assessing the presence and diversity of sociodemographic and behavioral factors related to Post COVID-19 Condition

PONE-D-25-08167R1

Dear Dr. Dolatabadi,

We’re pleased to inform you that your manuscript has been judged scientifically suitable for publication and will be formally accepted for publication once it meets all outstanding technical requirements.

Kind regards,

Issa Atoum

Academic Editor

PLOS ONE

Additional Editor Comments (optional):

Please ensure that Table 2 and all figures—particularly Figure 3—are properly aligned. The text in Figure 3 should have appropriate vertical spacing. Additionally, the abstract must not exceed the word limit.
---

## [Editor Report · Acceptance letter]

PONE-D-25-08167R1

PLOS ONE

Dear Dr. Dolatabadi,

I'm pleased to inform you that your manuscript has been deemed suitable for publication in PLOS ONE. Congratulations! Your manuscript is now being handed over to our production team.

Kind regards,

on behalf of

Dr. Issa Atoum

Academic Editor

PLOS ONE